# Pancreatic Cancer: *BRCA* Targeted Therapy and Beyond

**DOI:** 10.3390/cancers15112955

**Published:** 2023-05-28

**Authors:** Fergus Keane, Catherine A. O’Connor, Wungki Park, Thomas Seufferlein, Eileen M. O’Reilly

**Affiliations:** 1Department of Medicine, Memorial Sloan Kettering Cancer Center, New York, NY 10065, USA; keanef@mskcc.org (F.K.); oconnorc@mskcc.org (C.A.O.); parkw1@mskcc.org (W.P.); 2David M. Rubenstein Center for Pancreatic Cancer Research, New York, NY 10065, USA; 3Department of Medicine, Weill Cornell Medical College, New York, NY 10065, USA; 4Department of Internal Medicine, Ulm University Hospital, 89081 Ulm, Germany; thomas.seufferlein@uniklinik-ulm.de

**Keywords:** *BRCA*, pancreatic cancer, homologous recombination, PARP inhibitors, platinum, targeted therapy

## Abstract

**Simple Summary:**

Pancreatic cancer is associated with poor outcomes for several reasons, including diagnosis at an advanced stage, the absence of effective screening for the diagnosis, and resistance to treatments. Pancreatic cancers associated with *BRCA1/2* mutations have emerged as a distinct subgroup with sensitivity to other treatments and, in some cases, durable responses. Furthermore, beyond *BRCA1/2* mutations, there is increasing recognition that other gene mutations may behave in a similar manner. The focus of this review is to discuss recent developments in the management of *BRCA*-associated pancreatic cancer, emerging therapeutic strategies, and future directions for this subgroup of patients.

**Abstract:**

Pancreatic ductal adenocarcinoma (PDAC) is projected to become the second leading cause of cancer-related death in the US by 2030, despite accounting for only 5% of all cancer diagnoses. Germline g*BRCA1/2*-mutated PDAC represents a key subgroup with a favorable prognosis, due at least in part to additional approved and guideline-endorsed therapeutic options compared with an unselected PDAC cohort. The relatively recent incorporation of PARP inhibition into the treatment paradigm for such patients has resulted in renewed optimism for a biomarker-based approach to the management of this disease. However, g*BRCA1/2* represents a small subgroup of patients with PDAC, and efforts to extend the indication for PARPi beyond *BRCA1/2* mutations to patients with PDAC and other genomic alterations associated with deficient DNA damage repair (DDR) are ongoing, with several clinical trials underway. In addition, despite an array of approved therapeutic options for patients with *BRCA1/2*-associated PDAC, both primary and acquired resistance to platinum-based chemotherapies and PARPi presents a significant challenge in improving long-term outcomes. Herein, we review the current treatment landscape of PDAC for patients with *BRCA1/2* and other DDR gene mutations, experimental approaches under investigation or in development, and future directions.

## 1. Introduction

By 2030, pancreatic ductal adenocarcinoma (PDAC) is projected to become the second-leading cause of cancer-related mortality in the US [1]. The 5-year survival rate for PDAC is increasing and is now 12%, up from 5–6% in the previous decade [2]. Approximately 80% of new diagnoses are in the setting of advanced or metastatic disease [3], and in those patients who do undergo curative-intent surgery, disease recurrence is observed in up to 80% [4]. Until recently, precision medicine approaches to the management of PDAC have demonstrated modest incremental benefits; however, recent developments have given cause for renewed hope for specific subsets of patients.

The Pancreas Olaparib Ongoing (POLO) trial was the first phase III trial to establish a biomarker-based approach to the management of PDAC for patients with germline g*BRCA1/2* mutations [5,6], highlighting *BRCA*-associated PDAC as a distinct biological group with the potential for a personalized approach to treatment. Stadler et al. reported a pan-cancer analysis of almost 12,000 patients at Memorial Sloan Kettering Cancer Center (MSK) across over 50 tumor types who underwent comprehensive germline testing [7]. The OncoKB Precision Oncology Knowledge Base [8] was used to define the actionability of identified gene alterations, and 19.6% of PDAC cases were identified to harbor a pathogenic or likely pathogenic germline variant. Earlier, Lowery and colleagues carried out comprehensive germline testing on 615 patients with exocrine pancreatic neoplasms and identified pathogenic germline variants in 19.8% [9]. In both studies, pathogenic or likely pathogenic variants in *BRCA1/2* and other genes associated with homologous recombination accounted for a substantial proportion of the variants identified. With the National Comprehensive Cancer Network (NCCN) [10] and American Society of Clinical Oncology (ASCO) guidelines [11] recommending both germline and somatic testing for PDAC, there is potential for the incorporation of precision medicine approaches in a greater number of patients with PDAC than before. Herein, we review recent advances and future directions in the approach to the management of patients with PDAC, with a focus on *BRCA1/2* and other key HR gene mutations.

## 2. *BRCA* and DNA Damage Repair Pathways

Based on the type and frequency of structural variations identified by whole exome sequencing of a large cohort of PDAC tumor samples, Waddell et al. proposed four distinct PDAC subtypes: stable, locally rearranged, scattered, and unstable [12]. The unstable subtype is characterized by recurrent structural variation events owing to germline and somatic mutations of genes known to be involved in DNA damage repair (DDR), including *BRCA1, BRCA2, PALB2, RAD51C, RAD51D,* and *ATM*. These genes are relevant for the normal function of homologous recombination (HR) repair. Germline and somatic alterations in these key genes are now recognized to potentially lead to a HR deficient (HRD) phenotype within a tumor [13], which results in deficient double-stranded DNA break repair [14,15]. The roles of the *BRCA1* and *BRCA2* genes in encoding proteins required for HR are well described [16]. More recently, the function of *PALB2* as a “binding protein” for *BRCA2*, localizing *BRCA2* to sites of DNA breaks [17], has also been well supported. Together, *BRCA1/2* and *PALB2* are recognized as “core” genes that, when mutated, confer HRD in most settings of PDAC. Other “non-core” DDR variants and their role in inducing HR are less well defined but of growing interest.

### 2.1. Prevalence of BRCA and HRD in Pancreatic Cancer

The initial analysis of the Know Your Tumor registry trial identified mutations in the HR-DDR (DNA damage repair) pathway in 25% of 820 patients with PDAC, subdivided into three groups: Group 1: *BRCA1/2* and *PALB2*; Group 2: *ATM/ATR/ATRX*; and Group 3: *BAP1, BARD1, BRIP1, CHEK1/2, RAD50/51/51B*, or *FANCA/C/D2/E/F/G/L* [18]. While g*BRCA1/2* and g*PALB2* mutations are identified in 5–6% of an unselected PDAC population [14], rates of g*BRCA* mutations are known to be enriched in certain populations, including those of Ashkenazi Jewish heritage (5–16%), patients with a family history of pancreatic cancer (5–19%), and those with a family history of ovarian or breast cancer (5–10%) [19,20,21,22]. A retrospective analysis of the first 2206 patients screened for enrollment in the POLO trial demonstrated an overall prevalence of *BRCA1/2* mutations of 7.2% (5.8% after exclusion of populations known to be enriched with people of Ashkenazi Jewish heritage) and identified substantial geographic as well as some racial variability in the prevalence of g*BRCA* mutations amongst those screened, acknowledging a degree of selection bias given that almost 20% of patients enrolled in this trial had a prior documented g*BRCA* mutation [23].

While most patients with PDAC and an associated HRD phenotype have a pathogenic germline variant, a minority (2–4%) harbor a somatic-only HRD alteration. The widespread use of next-generation sequencing (NGS), as well as improvements in the sequencing assays utilized, has permitted more comprehensive identification of alterations in DDR genes. Waddell et al.’s whole genome sequencing and copy number variation analysis on 100 patients with PDAC identified two with a somatic *BRCA1* mutation and three with a somatic *BRCA2* alteration [12]. Work from our group demonstrated somatic HRD mutations in 4% of our cohort with PDAC and observed similarly favorable outcomes relative to those with germline HRD who received first-line platinum-based chemotherapy [14].

### 2.2. BRCAness Phenotype

While pathogenic variants in the DDR pathway have the potential to result in HRD and impaired double-stranded DNA break repair, as exemplified by *BRCA1/2* mutations, this does not appear to be the case for all DDR pathogenic variants. The concept of *BRCA*ness or HRD phenotype, defined as double-strand break repair deficiency in the absence of a *BRCA1/2* variant [24,25], is of growing interest across tumor types, including PDAC. Tumors exhibiting *BRCA*ness appear to respond favorably to DNA-damaging therapies such as platinum-based chemotherapy and poly-ADP ribose polymerase inhibitors (PARPi). There is an increasing recognition that, beyond focused germline and somatic profiling of HR-associated genes, evaluating genomic signatures through validated HRD scores may represent a more inclusive method of determining tumors that may respond to DDR-targeted treatments. These methods are more established in the context of breast and ovarian cancers [26,27]. Candidate variants of particular interest in PDAC and whether they confer a *BRCA*ness phenotype include *PALB2, RAD51, RAD50, CHEK2, ATM, BRIP,* and *BLM* variants.

Several methods for determining HRD status are in various stages of development. DNA “scars”, referring to gross chromosomal abnormalities and mutational signatures, are specific genomic features that are characteristic of HRD [28]. The three key types of chromosomal aberrations associated with HRD are telomeric allelic imbalance (TAI), large-scale state transitions (LSTs), and loss of heterozygosity (LOH) [29,30,31]. In triple-negative breast cancer, a composite score of TAI, LST, and LOH is a robust biomarker for identification of HRD status, demonstrating improved pathologic complete response rates to platinum-based chemotherapy in those with a high combined score, regardless of *BRCA1/2* status [26]. This strategy of HRD detection is the basis of Myriad’s MyChoice HRD assay, which, in addition to testing for pathogenic *BRCA1/2* mutations, also evaluates Genomic Instability Scores (GIS) based on TAI, LST, and LOH and is now an FDA-approved companion test to determine eligibility for olaparib and niraparib [32,33].

Beyond genomic scars, stereotyped mutational patterns based on the accumulation of single-base substitutions (SBS), insertion deletions (indels), and rearrangements are observed in tumors with HRD [28,34]. Signature 3, a SBS pattern associated with large deletions and microhomology, is observed regularly in *BRCA1/2* and *PALB2*-mutated PDAC [12,35] and has been demonstrated to predict response to platinum-based chemotherapy in patients with PDAC [15]. Other tests based on NGS have been developed, most notably HRDetect, which employs a weighted model of mutational signatures based on whole genome sequencing to detect HRD in breast, ovarian, and pancreatic cancer [36]. A recent meta-analysis of over 60 studies and >21,000 patients with PDAC identified the prevalence of HRD to be 14.5–16.5% based on targeted NGS but 24–44% by whole genome or whole exome sequencing [37], highlighting the need to clarify the definition and optimal method of detection of HRD in PDAC.

In one large study of 391 patients with PDAC, including 49 with g*BRCA1/2* or g*PALB2* pathogenic variants, HRD classifiers, including (i) GIS using Myriad’s MyChoice assay, (ii) Signature 3, (iii) HRDetect, and (iv) structural variant burden, were applied [15]. In this study, GIS scores had a sensitivity of 91% and a specificity of 83% for the identification of HRD, and improved clinical outcomes with platinum chemotherapy were associated with higher GIS scores. Most notably, HRDetect appeared to outperform both Sig3 and GIS scores in its ability to identify HRD PDAC, with high HRDetect scores present in up to 10% of patients who did not have a germline HR gene mutation. A HRDetect score >0.7 predicted g*BRCA1/PALB2* deficiency with a sensitivity of 98% and specificity of 100%. In addition, high HRDetect scores predicted improved survival in patients receiving platinum-based chemotherapy. A notable feature of this study was that pathogenic variants in *CHEK2* and *ATM* (*N* = 2 and *N* = 6, respectively) did not result in HRD by any of the above criteria. Identifying which, if any, of the DDR gene mutations result in a HRD phenotype and would thus present similar therapeutic actionability as *BRCA1/2* mutations is of considerable interest. As an example, our group at MSK recently evaluated 46 patients with PDAC and germline or somatic *ATM* variants [38] and determined that while *ATM*-mutated PDAC is associated with a favorable overall survival (OS) relative to genomically unselected PDAC, pathogenic variants in *ATM* did not appear to confer a HRD signature [38]. Several studies are underway evaluating other candidate HRD genes and whether they confer a HRD phenotype, as well as additional HRD/*BRCA*ness detection methods, including integrating DNA- and RNA-based HRD detection methods together [39]. The selected tests for evaluating HRD status are summarized in Table 1.

## 3. Treatment of *BRCA* and HRD-Mutated Pancreatic Cancer

### 3.1. Platinum-Based Chemotherapy

Platinum chemotherapies can crosslink purine bases in DNA, interrupting DNA transcription and replication, which leads to the accumulation of DNA damage due to a limited DNA damage repair capacity. The body of evidence supporting the use of platinum-based therapy in patients harboring *BRCA1/2* mutations is ever-growing, including in PDAC, with several studies demonstrating superior outcomes with platinum-based chemotherapy in patients with *BRCA1/2* mutations compared with unselected PDAC cohorts. Data suggests that the pleiotropic effects of mutant *BRCA1/2* are tumor-lineage dependent and that the therapeutic relevance of g*BRCA1/2* mutations is of most relevance in “*BRCA*-associated cancer types”, namely pancreatic, breast, ovarian, and prostate cancer [40].

In the Know Your Tumor Program, for patients with advanced PDAC who received platinum-based chemotherapy, the mOS was 1.27 years for HR-DDR mutated patients (*N* = 53) versus 1.45 years for patients with proficient HR-DDR status (*N* = 258), representing a statistically significant and clinically meaningful difference (*p* = 0.000072; HR 0.44; 95% CR 0.29 to 0.66). Data from our group at MSK demonstrated that in 50 patients with advanced PDAC and a germline or somatic HR-associated mutation treated with systemic therapy, a median progression-free survival (mPFS) advantage was observed for those in receipt of a platinum regimen versus a non-platinum regimen (12.6 vs. 4.4 months, HR 0.44, 96% CI 0.29–0.67, *p* < 0.01) [14]. This analysis also demonstrated that patients with biallelic HR mutations had higher genomic instability and derived further benefit from front-line platinum therapy, with a significantly improved mPFS of 13.3 months (95% CI, 0.26–0.70) in platinum-treated patients versus 3.8 months (95% CI, 2.79—not reached (NR); *p* < 0.0001) in non-platinum-treated patients. Momtaz et al. reported on a large cohort of patients with PDAC and germline or somatic *BRCA1/2* mutations [41]. Of 81 patients with stage IV disease, the mOS for patients who received upfront platinum-based therapy (*N* = 65) was 23 months (95% CI, 19–26), versus 29 months (95% CI, 19 months to NR) for those who did not receive frontline platinum-based therapy (*N* = 14). Of the 14 patients who did not receive platinum-based therapy in the first line, 10 did receive it in the second line, and favorable responses were noted (a PR was noted in 7 patients and SD in 1 patient as the best response), with a median duration of therapy of 11 months (range, 1–35). Notably, amongst those with metastatic disease who did receive front-line platinum-based therapy, the mOS for patients with biallelic status (*N* = 39) was 26 months (95% CI, 20–52 months) and 8.66 months (95% CI, 6.2—NR) for those with monoallelic status (*N* = 4). Beyond the advanced PDAC setting, several studies have also demonstrated an advantage associated with the receipt of platinum-based therapy in the neoadjuvant setting for patients with *BRCA1/2*-mutated PDAC [42,43].

To enhance response and potentially delay resistance, O’Reilly et al. evaluated the role of veliparib in combination with cisplatin and gemcitabine in untreated metastatic and locally advanced PDAC with *BRCA1/2* or *PALB2* mutations [44]. Patients were randomized to receive cisplatin and gemcitabine, with or without veliparib. Delayed emergence of resistance and improved survival were not observed with the addition of veliparib; however, high response rates in both groups were observed (74.1% in the experimental arm and 65.2% in the control arm, *p* = 0.55), as well as a favorable OS signal from first-line cisplatin-based therapy in both arms (15.5 months and 16.4 months, respectively, *p* = 0.73), establishing cisplatin and gemcitabine as a standard approach in g*BRCA1/2*- and *PALB2*- mutated PDAC and an alternative to FOLFIRINOX.

### 3.2. Poly-ADP Ribose Polymerase Inhibitors

The PARP enzymes have a key role in base excision repair of single-stranded DNA breaks [45], and inhibition of these enzymes results in the accumulation of single-stranded breaks (SSBs), leading to replication fork collapse, and the generation of double-stranded breaks (DSBs). As discussed earlier, DSBs rely on HRR for repair and to avoid cell cycle arrest and apoptosis [46]. Thus, in patients with mutations in HRR genes, PARPi can be employed to create a state of synthetic lethality in tumors [46]. Select recently-completed and ongoing trials evaluating the role of PARPi for *BRCA*-mutated and HRD PDAC are summarized in Table 2 and Table 3. In a multicenter phase 2 study of patients with breast, prostate, ovarian, and PDAC tumors with g*BRCA1/2* mutations, the use of oral PARPi olaparib demonstrated an overall response rate of 26.2% (95% CA 21.3–31.6) across all patients [47]. Specifically, in the PDAC cohort, which included 23 patients, all with advanced disease, an overall response rate of 21.7% was observed, including one complete response and four partial responses.

Olaparib was prospectively evaluated in the phase III Pancreas Olaparib Ongoing (POLO) trial, which enrolled 154 patients with g*BRCA1/2* mutations and advanced or metastatic PDAC [5,6]. Patients were required to have demonstrated disease response or stability following a minimum of 16 weeks of platinum-based chemotherapy. The primary end point of PFS was met, with a mPFS of 7.4 months in the olaparib arm compared with 3.8 months in the placebo arm (HR 0.53, *p* = 0.004). Following this data, FDA approval for olaparib was granted in December 2019. The secondary endpoint of OS, however, was not met (median 19.0 months versus 19.2 months; HR 0.83, 95% CI, 0.56–1.22, *p* = 0.3487). OS at 36 months was 33.9% for olaparib and 17.8% for placebo, and mPFS2, defined as median time from randomization to second disease progression, was 16.9 months for olaparib versus 9.3 months for placebo (hazard ratio, 0.66; *p* = 0.0613), indicating a trend toward benefit for olaparib, though not reaching statistical significance. Notably, quality of life scores between the arms were equivalent. POLO was the first trial to establish a biomarker-based approach to the management of PDAC in patients with g*BRCA* mutations; however, the lack of OS advantage and the placebo control arm have led to questions regarding the magnitude and importance of the observed PFS benefit.

Reiss et al. conducted a phase 2 trial evaluating PARPi rucaparib in the post-platinum maintenance setting in patients with PDAC harboring germline or somatic *BRCA1/2* or *PALB2* alterations [48]. They reported a mPFS of 13.1 months (95% CI, 4.4–21.8) and a mOS of 23.5 months (95% CI, 20–27). An ORR of 41.7% was observed, which included 3 complete responses and 12 partial responses. Rucaparib was also evaluated in the phase 2 RUCAPANC study, which included patients with germline or somatic *BRCA1/2* mutations [49]. While two-thirds of patients had received prior platinum, platinum sensitivity and prior platinum were not mandated. The study was terminated early based on a modest ORR of 15.8%, and notably, none of the patients who had platinum-resistant disease at the time of enrollment had an objective response. Indeed, sensitivity to platinum has emerged as a key predictive biomarker for sensitivity to PARPi.

In a phase 2 study of veliparib in 16 patients with metastatic or locally advanced PDAC and g*BRCA1/2* or *PALB2* mutations, Lowery et al. reported no objective response, although stable disease for >4 months was noted in four patients [50]. Most patients enrolled in this study had platinum-resistant PDAC, which most likely explains the limited signal observed. Beyond g*BRCA* mutations, Javle and colleagues described the clinical outcomes of olaparib monotherapy in patients with PDAC and alterations in other DDR genes, including *ATM*, *PALB2*, *ARID1A*, s*BRCA*, *PTEN*, *RAD51*, *CCNE*, and *FANCB*. They identified that patients with platinum-sensitive PDAC had an improved mPFS (4.1 months, 95% CI, 3.6–7.8) over those with platinum-resistant PDAC (2.2 months, 95% CI, from 1.8 to not reached, *p* = 0.01) [51]. This benefit was also observed in the analysis of OS (10.5 vs. 5.4 months for platinum-sensitive versus platinum-resistant disease, respectively, *p* = 0.03). Based on this study, in addition to others, it is apparent that in PDAC, the role of PARPi is maximized in the platinum-sensitive maintenance setting.

Beyond their role in advanced disease, the utility of PARPi in the early-stage setting is under investigation in clinical trials. The APOLLO study (NCT04858334) is a randomized phase II study of adjuvant olaparib versus placebo in patients with pancreatic cancer and germline or somatic *BRCA1/2* or *PALB2* mutations in whom all curative intent standard treatment has been completed [52].

### 3.3. Resistance to PARP Inhibition

Despite the rationale and enthusiasm for the use of PARPi in patients with *BRCA1/2* mutations (and other DDR gene alterations), primary resistance is observed in a notable proportion of patients, including those with platinum-sensitive PDAC. In the POLO trial, despite all patients enrolled having confirmed platinum sensitivity in the context of g*BRCA1/2* mutations, approximately one fifth of patients had evidence of progression of disease on first assessment [5,6]. Similarly, in the phase II study by Reiss et al. evaluating maintenance rucaparib in the post-platinum maintenance setting for patients with germline or somatic *BRCA1/2* or *PALB2* pathogenic variants, 16% of patients experienced progression of disease within the first 8 weeks of treatment [48]. This builds upon the data from Golan and colleagues, in which patients with PDAC and g*BRCA1/2* mutations had HRDetect performed on their pancreatic cancer tumor samples, and 12% did not have a HRD signature [15], which may account for primary resistance to PARPi in a proportion of patients.

Secondary acquired resistance to PARPi is common, and several mechanisms are postulated [53]. In several studies, hyperactivation of the ATR/CHK1 pathway, resulting in maintained genomic stability, has been suggested as a mechanism of PARPi resistance in *BRCA*-driven tumors [54,55], and forms the basis of several efforts to overcome resistance. This pathway is discussed in greater detail below. In addition, the emergence of reversion mutations, resulting in the restoration of HRR following treatment with PARPi, has been described. In a study by Pettitt et al., data from the literature pertaining to 308 reversion mutations in the setting of PARPi or platinum resistance identified in 91 patients were reported and suggest that for *BRCA2* mutations, the position of the mutation may affect the risk of reversion [56]. In addition, they concluded that many reversions are predicted to encode highly immunogenic neopeptides, which may provide a potential opportunity for avoiding resistance, for example, through the integration of immunotherapy. Specific to PDAC, preclinical models of *ATM*-deficient pancreatic cancer cells identified an association between the upregulation of alternative-end joining via upregulation of drug efflux transporters and detoxication enzymes and resistance to PARP inhibition [57]. However, the mechanisms of PARPi resistance in pancreatic cancer appear complex and, as of yet, are less well defined [58]. Ongoing efforts to deepen the responses to PARPi as well as to delay resistance are underway, with several combination strategies under investigation in PDAC (Table 2).

## 4. Combination Treatment Strategies

### 4.1. PARP Inhibition and Chemotherapy

As discussed earlier, our group evaluated veliparib in combination with cisplatin and gemcitabine in untreated metastatic and locally advanced PDAC with *BRCA1/2* or *PALB2* mutations in a randomized phase 2 study [43] and demonstrated that while a favorable overall survival signal from first-line cisplatin and gemcitabine chemotherapy (+/− veliparib) was observed in both arms (15.5 months and 16.4 months, respectively, *p* = 0.73), improved survival and delayed resistance were not observed with the addition of veliparib in this trial. The SWOG S1513 trial was a randomized phase 2 study that evaluated veliparib with or without FOLFIRI in the second line for patients with stage IV PDAC [59]. Accrual was halted early, with 123 patients enrolled, due to the lack of benefit from the addition of veliparib. For the entire group, mOS was not improved with the addition of veliparib (5.4 vs. 6.5 months, HR 1.23, *p* = 0.28). Notably, for patients with HR-DDR defects (*N* = 22, 19%) compared to those without HR-DDR gene defects, mPFS and mOS were 7.3 vs. 2.5 months (*p* = 0.05) and 10.1 vs. 5.9 months (*p* = 0.17), respectively, with FOLFIRI alone, and 2.0 vs. 2.1 months (*p* = 0.62) and 7.4 vs. 5.1 months (*p* = 0.10), respectively, with veliparib plus mFOLFIRI. Both arms performed favorably compared with the overall group, indicating a potential role for irinotecan-based therapy in patients with such defects. In addition, an ongoing non-randomized phase 2 study is evaluating 5-FU and liposomal irinotecan in combination with rucaparib in patients with metastatic PDAC with *BRCA1/2*, *PALB2,* or other DDR-HRD genomic alterations who have not received systemic therapy in the metastatic setting (NCT03337087).

### 4.2. PARPi and Immune Checkpoint Inhibitors

In pre-clinical studies, PARPi have been demonstrated to enhance cancer-associated immunosuppression and upregulate PD-L1 expression [60]. In addition, the combination of PARPi and immune checkpoint blockade has demonstrated an accumulation of tumor neoantigens and activation of interferon pathways, resulting in a synergistic antitumoral effect [61,62] in pre-clinical studies. The phase 1b/2 PARPVAX study of niraparib plus nivolumab and niraparib plus ipilimumab in patients with platinum-sensitive PDAC was evaluated and compared to a historical reference [63]. Subgroup analysis of patients with *BRCA1/2* or *PALB2* variants demonstrated a mPFS of 3.7 months and a mOS of 12.2 months in the niraparib and nivolumab arm, versus a prolonged mPFS of 10.4 months and a mOS of 38 months in the niraparib plus ipilimumab arm, suggesting that CTLA-4 inhibitors may potentially have an enhanced benefit in combination with PARPi compared with PD-1 inhibitors, and this signal warrants further investigation.

The non-randomized phase 2 POLAR study (NCT04666740) is evaluating the combination of maintenance pembrolizumab and olaparib in patients with metastatic pancreatic cancer and HR deficiency (or exceptional response to platinum) post-platinum-based chemotherapy. The SWOG S2001 trial (NCT04548752) is a randomized study evaluating olaparib with/without pembrolizumab, also in the post-platinum maintenance setting, in patients with metastatic pancreatic cancer and g*BRCA1/2* mutations. Several studies are evaluating PARPi and immunotherapy +/− other agents, for example, PD-1 inhibitor dostarlimab and PARPi niraparib in patients with advanced *BRCA*-mutated tumors, including PDAC (NCT04673448, NCT04493060), dostarlimab, niraparib, and radiotherapy (NCI04409002), PD-L1 inhibitor avelumab, PARPi talazoparib, and MEK inhibitor binimetinib in patients with locally advanced or metastatic RAS-mutated solid tumors (NCT03637491, closed), Beyond *BRCA1/2* variants, the MAZEPPA trial is evaluating PARPi of olaparib or kinase inhibitor selumetinib plus PD-1 inhibitor durvalumab in patients with metastatic PDAC and *BRCA*ness (NCT04348045).

### 4.3. PARPi and Anti-Angiogenic Agents

Agents inhibiting vascular epidermal growth factor (VEGF) families result in a hypoxic state and subsequent downregulation of HRR gene expression [64]. Series suggest that this hypoxic tumor microenvironment results in the functional inactivation of *RAD51* and *BRCA* without a genetic alteration in these genes, with a resulting *BRCA*ness phenotype [65,66]. Some promise has been observed in combining PARPi and VEGF inhibitors in both ovarian cancer and prostate cancer. A randomized phase 2 study of olaparib with or without the pan-VEGF inhibitor cediranib demonstrated a significant radiographic PFS advantage over olaparib alone in men with metastatic castrate-resistant prostate cancer [67]. In ovarian cancer, the phase 3 PAOLA-1/ENGOT-ov25 trial of olaparib and bevacizumab in the post-platinum maintenance setting demonstrated a significant PFS advantage in those with *BRCA* and non-*BRCA* HRD mutations [32]. Olaparib combined with the anti-VEGF agent cediranib is under investigation in advanced solid tumors, including PDAC, in an ongoing phase 2 study (NCT02498613).

### 4.4. PARPi and Other Agents

ATR kinase-mediated DDR pathways are thought to promote tumor survival during PARP inhibition, and in pre-clinical studies, the ATR-inhibitor AZD6738 (ceralasertib) in combination with olaparib has demonstrated synergistic tumor inhibition [68]. A phase 2 study is currently underway investigating the role of AZD6738 with or without olaparib in patients with advanced solid tumors, including PDAC, who harbor *ARID1A* expression or *ATM* loss/mutations (NCT03682289). In addition, pre-clinical data from one study demonstrated that cells resistant to PARP inhibition showed evidence of elevated RAS/MAP kinase signaling, and a signal for partial reversal with the addition of MEK inhibition was observed [69]. On this basis, PARPi and MEK inhibition are being investigated both in the neoadjuvant (NCT04005690) and advanced settings (NCT03637491).

### 4.5. Immune Checkpoint Blockade in BRCA and HRD Pancreatic Cancer

PDAC is considered a prototypical immunogenically “cold” tumor, owing to an inherently immunosuppressive, hypoxic tumor microenvironment as well as a dense surrounding stroma. Outside of the small proportion of patients with microsatellite-instable PDAC (approximately 1%), a role for immune checkpoint blockade has been challenging to support, with poor responses observed in a phase 2 study from our group [70]. However, greater genomic instability in patients with PDAC with biallelic loss of *BRCA1* and *BRCA2* has been demonstrated across several series [14,71], suggesting a possible role for the incorporation of immune checkpoint blockade in this subset of patients. Terraro et al. reported the results of a series of 12 patients with chemotherapy-refractory pancreaticobiliary cancers and pathogenic germline variants in HRD genes (specifically *BRCA1, RAD51C, ATM, BRCA2,* and *RAD51D*) who received dual checkpoint blockade with ipilimumab and nivolumab. Of the ten patients with PDAC (out of a total of 12; 10 PDAC, 1 cholangiocarcinoma, and 1 ampullary carcinoma), two had a complete response to therapy, one had a partial response, and two had stable disease [72], indicating a potential role for immune checkpoint blockade as a therapeutic strategy in *BRCA*-mutated and other HRD PDAC.

### 4.6. Targeting DNA Replication Stress

While much of the focus of novel therapeutic strategies for PDAC has been on targeting HR, defective DDR may result in “replication stress”, defined as the perturbation of error-free DNA replication and the slowing of DNA synthesis, resulting in genomic instability and oncogenic transformation [73]. The ATR kinase has an important role in the cellular response to replication stress [74]. Replication protein A (RPA) binds single-stranded DNA (ssDNA) and recruits ATR, which results in the downstream activation of ATR-interacting proteins and subsequently ATR and CHEK1 through phosphorylation [74]. ATR-CHEK1 activation results in s-G2 cell cycle arrest and activation of HR repair, as well as replication fork stabilization [73,74]. CHEK1 is negatively regulated by WEE1 and MYT1 via phosphorylation and results in the activation of WEE1 and degradation of CDC25A [75,76]. Attempts at targeting replication stress have led to the development of several compounds, many of which are in preclinical and clinical development, including ATR inhibitors (ATRi), CHEK1 inhibitors, and WEE1 inhibitors.

ATRi elimusertib was evaluated as a monotherapy in a phase 1 trial of 21 patients with advanced solid tumors (NCT03188965). An ORR of 19% (4 of 21) was observed, all 4 occurring in patients harboring alterations in or loss of *ATM* [77]. Intravenous ATRi berzosertib was evaluated as a monotherapy and in combination with carboplatin in 40 patients with advanced solid tumors (NCT02157792). One patient with metastatic colorectal cancer harboring *ATM* loss and an *ARID1A* mutation had a complete response; another patient with platinum-refractory and PARPi-resistant ovarian cancer harboring a g*BRCA*1 mutation had a partial response [78].

CHEK1/2i prexasertib has been evaluated as a monotherapy in two studies. In a phase 1 study (NCT01115790) of 45 patients with advanced solid tumors, 2 patients had a partial response, and 15 patients had stable disease as the best overall response, with a more favorable signal in patients with squamous histology. Thus, an expansion cohort, including patients with squamous histology only, was carried out and demonstrated partial responses in 15% of patients with anal cancer and 5% of patients with head and neck tumors [79]. In the second study (NCT02203513), a phase 2 study, 42 patients with ovarian cancer were enrolled. Of 24 patients without *BRCA1/2* mutations, an ORR of 33% was observed, and among 18 patients harboring *BRCA1/2* mutations, an ORR of 11% was observed [80,81]. WEE1i adavosertib was evaluated in combination with chemotherapy (gemcitabine, paclitaxel, carboplatin, or pegylated liposomal doxorubicin) in a phase II study in 94 patients with primary platinum-resistant gynecologic malignancies (NCT02272790) [82]. A promising ORR of 32% was observed. ZN-C3 has been evaluated in a phase 1 study (NCT04158336) of patients with high-grade serous endometrial carcinoma and resulted in a complete response in one patient and two partial responses in eleven patients in total [83]. Several studies combining chemotherapy with inhibitors of the pathway are underway, including gemcitabine, carboplatin, and berzosertib in advanced ovarian cancer (NCT02627443), and gemcitabine, nab-paclitaxel, and adavosertib in PDAC (not specific to patients with HRD) (NCT02194829—active, not recruiting).

### 4.7. Other Promising Strategies in BRCA and HRD

Inhibitors of polymerase theta (POLθ, which is encoded by POLQ) are gaining interest as an additional therapeutic strategy in HRD malignancies. POLθ is the key enzyme in microhomology-mediated end-joining (MMEJ), and several studies have demonstrated the potential of POLθ inhibitors in evasion of MMEJ and inducing synthetic lethality [84,85]. In a key recent study in *BRCA*-mutated breast cancer and PDAC cell lines, either novobiocin, an inhibitor of the POLθ ATPase domain, or ART558, an inhibitor of the POLθ polymerase domain, demonstrated activation of the cGAS/STING pathway. Activation of this pathway drives the expression of type I interferon response elements, including PD-L1, and increases CD8+ T-cell tumor infiltration and activation, as well as the activation of antigen-presenting dendritic cells [86]. In addition, the antitumor activity of novobiocin was augmented with the addition of PD-1 blockade in a *BRCA2*-deficient mouse model in this study. The first human study of novobiocin will begin accruing patients with tumors with alterations in DNA repair genes in 2023 (NCT05687110). Current and emergent targeted strategies are summarized in Figure 1.

## 5. Conclusions and Future Directions

With modest progress in survival outcomes for patients with PDAC over the last two decades, the emergence of a personalized, biomarker-based approach to the management of PDAC is promising. Patients with *BRCA1/2* mutations represent a biologically distinct subgroup with a favorable predictive profile compared with an unselected PDAC population. In addition, patients with HRD status are emerging as a group that may also benefit from the incorporation of a similar personalized approach. However, despite developments in the management of patients with *BRCA1/2*- and HRD-associated PDAC, resistance is a critical limitation. Future efforts should focus on the continued development of accurate HRD detection methods. In addition, well-designed, adequately powered, randomized clinical trials of novel combination approaches to deepen responses and overcome resistance are imperative. Lastly, the incorporation of translational, interdisciplinary science and, most importantly, universal patient access to somatic and germline testing as recommended by international guidelines should be prioritized.

## Figures and Tables

**Figure 1 cancers-15-02955-f001:**
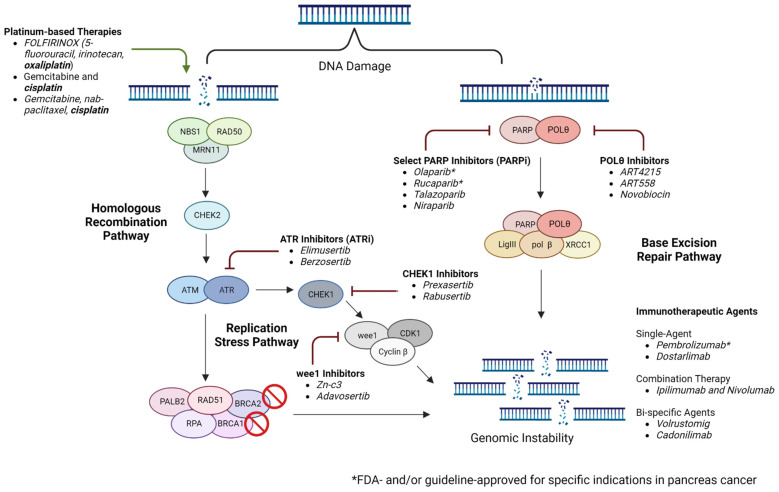
Targeting the DNA damage repair pathway in pancreatic cancer (created using Biorender.com accessed on 2 April 2023).

**Table 1 cancers-15-02955-t001:** Select Tests to Evaluate for HRD (not specific to pancreatic cancer).

Test	Description
Telomeric Allelic Imbalance (TAI) [28]	Allelic imbalance at the telomere of the chromosome is due to the propensity for inappropriate end joining in HRD, identified by single nucleotide polymorphism (SNP) genotyping.
Large Scale Transitions (LST) [29]	Chromosomal breaks larger than 10 Mb, which arise in HRD cells secondary to erroneous recombination between segments of the chromosome, are identified by single nucleotide polymorphism (SNP) genotyping.
Loss of Heterozygosity (LOH) [30]	Uniparental disomy is owing to inaccurate repair of sister chromatids during S/G2 phase, resulting in the loss of entire genes and the surrounding chromosomal region, as identified by single nucleotide polymorphism (SNP) genotyping.
Genomic Instability Score eg by Myriad Genetics MyChoice Assay	TAI + LST + LOH
Signature 3 (Sig 3) [35]	A single base substitution mutational pattern, associated with microhomology and large deletions, was identified by whole exome sequencing.
HRDetect [36]	A weighted model incorporating a weighted score of base substitution/rearrangement signatures, microhomology-mediated deletions, and an HRD score based on genomic scars identified by whole exome sequencing.

**Table 2 cancers-15-02955-t002:** Select Recent Completed Trials in *BRCA* and HRD PDAC.

NCT Identifier	Phase	Target Population	Experimental Arm	Control Arm	Primary Outcome	Other Key Findings
NCT02184195(POLO)	3	Metastatic PDAC, g*BRCA1/2* mutated, with platinum sensitivity	Olaparib	Placebo	PFS 7.4 mo (experimental arm) vs. 3.8 mo (control arm)	PFS 2 16.9 mo (experimental arm) vs. 9.3 mo (placebo arm). Quality of life scores equivalent. OS 19.0 mo (olaparib) vs. 19.2 months (placebo)
NCT03140670	2	Metastatic PDAC, s/*g BRCA1/2* or *PALB2* mutated, with platinum sensitivity	Rucaparib	N/A	PFS6 59.5%	mPFS 13.1 mo mOS 23.5 mo ORR 41.7%
NCT02042378 (RUCAPANC)	2	Metastatic or advanced PDAC, s/g *BRCA* mutation, 1–2 lines of previous treatment	Rucaparib	N/A	RR 15.8% (1 CR, 2 PR)	DCR 31.6% Terminated early due to insufficient response rate
NCT02184195	2	Metastatic or locally advanced PDAC, previously treated, with g*BRCA1/2* or *gPALB2* mutations	Cisplatin, gemcitabine (Arm B) plus veliparib (Arm A)	Cisplatin, gemcitabine (Arm B)	RR 74.1% for Arm A, 65.2% for Arm B	DCR 100% for Arm A, 78.3% for Arm B. mPFS 10.1 mo for Arm A, 9.7mo for Arm B. mOS 15.5 mo for Arm A, 16.4 mo for Arm B
NCT03404960	2	Locally advanced or metastatic PDAC with platinum sensitivity, in the platinum-sensitive, maintenance setting	Niraparib plus nivolumab and niraparib plus ipilimumab	N/A	6m PFS 20.6% in niraparib + nivo arm versus 59.6% in the niraparib + ipi arm	mOS 10.2 mo in niraparib + nivo, 38 mo for niraparib + ipi Higher grade 3 toxicity with niraparib + ipilimumab
NCT02184195	2	Metastatic or locally advanced PDAC, g*BRCA1/2* or *gPALB2*, 1–2 lines of previous treatment	Veliparib 300mg twice daily or veliparib 400mg twice daily	N/A	ORR 0%	mPFS 1.7 mo OS 3.1 mo
NCT0129673	1	Unresectable PDAC, not confined to those with *BRCA* or HRD mutations	Olaparib, cisplatin, irinotecan, and mitomycin	N/A		ORR 23% Grade 3 AEs in 89% 11% pts developed MDS 1 pt with g*BRCA2* had durable response for >4 years

Abbreviations: PDAC: pancreatic ductal adenocarcinoma; g*BRCA*: germline *BRCA*; DLT: dose-limiting toxicities; mPFS: median progression-free survival; AE: adverse events; mo: months; ORR: objective response rate; s/g: somatic or germline; N/A: not applicable; PFS6: progression-free survival at six months; mOS: median overall survival; ORR: overall response rate; CR: complete response; GnP: gemcitabine plus nab-paclitaxel; PR: partial response; mFFX: modified FOLFIRINOX; SD: stable disease; DCR: disease control rate; POD: progression of disease; MDS: myelodysplastic syndrome.

**Table 3 cancers-15-02955-t003:** Select ongoing trials for *BRCA*, *BRCA*ness, and HRD pancreatic cancer.

NCT Identifier	Phase	Target Population	Experimental Arm	Control Arm	Primary Outcome	Anticipated Completion
NCT02677038	2	Metastatic PDAC with s/gHRD gene mutations (except *BRCA1/2*) or a family history suggestive of a HRD mutation.	Olaparib	N/A	ORR	November 2022
NCT04858334 (APOLLO)	2	Surgically removed PDAC, post-adjuvant chemo(radiotherapy), and a g/s*BRCA1/2* or *PALB2* mutation.	Olaparib	N/A	RFS relapse free survival	November 2024
NCT04666740 (POLAR)	2	Metastatic PDAC with core or non-core HRD mutations or platinum sensitivity.	Olaparib plus pembrolizumab	N/A	PFS	January 2024
NCT04548752	2	Metastatic PDAC with g*BRCA1/2* mutations.	Olaparib plus pembrolizumab	Olaparib	PFS	March 2025
NCT04171700 (LODESTAR)	2	Metastatic solid tumors, including PDA with mutations in *BRCA1/2, PALB2, RAD51C, RAD51D, BARD1, FANCA, NBN, RAD51,* or *RAD51B.*	Rucaparib	N/A	ORR	March 2023
NCT03553004 (NIRAPANC)	2	Locally advanced or metastatic PDAC with g/sHRD mutations.	Niraparib	N/A	ORR	February 2025
NCT03601923	2	Locally advanced or metastatic PDAC with g/s*BRCA1/2, PALB2, CHEK2,* and *ATM* mutations.	Niraparib	N/A	PFS	February 2025
NCT04493060	2	Metastatic PDAC with a *BRCA1/2* or *PALB2* mutation.	Niraparib plus dostarlimab	N/A	DCR12	September 2022
NCT04673448	1	Metastatic solid tumors, including PDAC with a g/s*BRCA1/2* mutation.	Niraparib plus TSR-042	N/A	PFS	March 2026
NCT03601923	2	Previously treated PDAC with g/sHRD mutations.	Niraparib plus small field palliative radiation	N/A	PFS	February 2025
NCT04005690	1	Locally advanced or metastatic PDAC.	Arm 1: cobimetinib Arm 2: olaparib	N/A	Feasibility of obtaining tumor tissue pre and post treatment	February 2025
NCT04550494	2	Locally advanced or metastatic solid tumors, including PDAC with g/s DDR gene alterations.	Talazoparib	N/A	PD	August 2022
NCT04348045 MAZEPPA)	2	Metastatic PDAC and known somatic profile—stratified by *BRCA*ness and *KRAS* mutation status.	Pts with *BRCA*ness (somatic profile): olaparib (Arm A) Pts without *BRCA*ness and with KRAS mutation are randomized to durvalumab + selumetinib (Arm B) or FOLFIRI (Arm C)	N/A	PFS	December 2024
NCT03337087	1/2	Metastatic PDAC (colorectal, gastroesophageal, and biliary cancer) and for the phase 2 component, *BRCA1/2, PALB2,* or HRD, untreated in the metastatic setting.	Liposomal irinotecan, fluorouracil, calcium leucovorin, and rucaparib	N/A	DLT rate (Phase 1) ORR (Phase 1b) Best response (Phase 2)	Not recruiting
NCT02194829	1	Metastatic or unresectable PDAC with alterations in DDR genes, previously untreated.	Novobiocin	N/A	MTD RP2D	Anticipated to open in 2023

Abbreviations: AE: adverse events; PDAC: pancreatic ductal adenocarcinoma; DCR12: disease control rate at 12 weeks; ORR: overall response rate; DLT: dose limiting toxicities; HRD: homologous recombination deficiency; N/A: not applicable; DDR: DNA damage repair; MTD: maximum tolerated dose; PFS: progression free survival; GnP: Gemcitabine/Nab-Paclitaxel; OS: overall survival; mFFX: modified FOLFIRINOX; PK: pharmacokinetics; ctDNA: circulating tumor DNA; RP2D: recommended phase 2 dose.

## Data Availability

No new data were created or analyzed in this study. Data sharing is not applicable to this article.

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
