# Peer review of "Pancreatic Cancer: BRCA Targeted Therapy and Beyond"

_cancers, 2023, doi:10.3390/cancers15112955_

Round 1
Reviewer 1 Report
The author made a comprehensive review on the treatment of pancreas ductal adenocarcinoma patients with BRCA1/2 and other DDR gene mutations by focusing on its recent therapeutic methods like platinum-based chemotherapy, PARP inhibitors and combination treatment strategies. Although such topic has been well reviewed recently by Marciano ND, et al. (BRCA-Mutated Pancreatic Cancer: From Discovery to Novel Treatment Paradigms. Cancers (Basel). 2022 May 16;14(10):2453. doi: 10.3390/cancers14102453), this manuscript provides some useful knowledge renewal especially on details of available clinical trials. I would thus propose to consider it for publication.
Comments:
1. The subtitle of “Resistance to PARP Inhibition” presents twice in lines 285 and 313, which should be edited.
2. Typo for “bi-omarker-based” in line 55.
Overall, it is well organized.
Author Response
Sincere thanks for your kind comments, and for your helpful suggestions.
Comment: The subtitle of “Resistance to PARP Inhibition” presents twice in lines 285 and 313, which should be edited.
Response: Thank you, we have edited this as suggested.
Comment: Typo for “bi-omarker-based” in line 55.
Response: Thank you, we have edited this as suggested.
Reviewer 2 Report
The authors did an excellent job in this review. Pancreatic ductal adenocarcinoma (PDAC) is a deadly cancer that is anticipated as the second leading cause of cancer-related death in the United States by 2030. Germline (g)BRCA1/2-mutated PDAC is a subgroup of patients with a favourable prognosis, due to additional approved and guideline-endorsed therapeutic options. However, gBRCA1/2 represent a small subgroup of patients with PDAC, and efforts to extend the indication for PARPi beyond BRCA1/2 mutations are ongoing. PARP inhibition is a relatively new treatment for PDAC that has resulted in renewed optimism for a biomarker-based approach in the management of this disease. The authors are very well aware of primary and acquired resistance to platinum-based chemotherapies and PARPi presents a significant challenge in improving long-term outcomes. They way the authors presented this very well. Herein, they discuss the current treatment landscape of PDAC for patients with BRCA1/2 and other DDR gene mutations, experimental approaches under investigation and in development, and future directions. With modest progress in survival outcomes for patients with PDAC over the last two decades, the emergence of a personalized, biomarker-based approach to the management of PDAC is promising. Patients with BRCA1/2 mutations represent a biologically distinct subgroup, with a favourable predictive profile compared with an unselected PDAC population. In addition, patients with HRD status are emerging as a group who may also benefit from the incorporation of a similar personalized approach. However, despite developments in the management of patients with BRCA1/2- and HRD-associated PDAC, resistance is a critical limitation. Future efforts should focus on the continued development of accurate HRD detection methods. In addition, well-designed, adequately powered, randomized clinical trials of novel combination approaches to deepen responses and overcome resistance are imperative. Lastly, the incorporation of translational, interdisciplinary science, and most importantly, universal patient access to somatic and germline testing as recommended by international guidelines should be prioritized. Please add the full form of all the abbreviations.Good
Author Response
Thank you very much for your kind comments, and helpful suggestions.
Comment: Please add the full form of all the abbreviations.
Response: Thank you, we have now added the full form of all abbreviations.
Reviewer 3 Report
This is a comprehensive review of BRCA and associated mutatoins in PDAC.
1. The paper gives wide range of information from diagnosis to tretment. It would be great if the authors can provide some flowchart from diagnosis of BRCA and HRD to treatment selection.
2. There are various tests to evaluate BRCA mutations and HRD. Can the authors provide a table showing necessary samples, results and cost etc. for each test?
Author Response
Thank you for your comments, and we are very grateful for your thoughtful comments which we believe enhance this manuscript.
Comment: The paper gives wide range of information from diagnosis to tretment. It would be great if the authors can provide some flowchart from diagnosis of BRCA and HRD to treatment selection.
Response: Thank you for this suggestion. As the majority of the treatments discussed are approved only in patients with germline BRCA1/2 or PALB2 mutations, and not for patients with HRD, instead of providing a flowsheet, we have indicated in Figure 1 which treatments are approved and/or guideline-endorsed for patients, and hope this provides clarity for readers.
Comment: There are various tests to evaluate BRCA mutations and HRD. Can the authors provide a table showing necessary samples, results and cost etc. for each test?
Response: Thank you. We have addressed this suggestion by adding an additional table (Table 3), including details of the tests available, samples required and the results yielded. We have not included costs at this time as these are variable depending on location, and costs change regularly.
Thanks again for your helpful suggestions which we hope we have addressed adequately.
Round 2
Reviewer 1 Report
Ready for publication
Reviewer 2 Report
Revised version is ready to accept in present form.
Reviewer 3 Report
I have no further comments.